# Mitochondrial Dysfunction and Coenzyme Q10 Supplementation in Post-Viral Fatigue Syndrome: An Overview

**DOI:** 10.3390/ijms25010574

**Published:** 2024-01-01

**Authors:** David Mantle, Iain Parry Hargreaves, Joan Carles Domingo, Jesus Castro-Marrero

**Affiliations:** 1Pharma Nord (UK) Ltd., Morpeth, Northumberland NE61 2DB, UK; 2School of Pharmacy and Biomolecular Sciences, Liverpool John Moores University, Liverpool L3 3AF, UK; i.hargreaves@acl.ac.uk; 3Department of Biochemistry and Molecular Biomedicine, Faculty of Biology, University of Barcelona, 08028 Barcelona, Spain; jcdomingo@ub.edu; 4Research Unit in ME/CFS and Long COVID, Rheumatology Division, Vall d’Hebron Research Institute, Universitat Autònoma de Barcelona, 08035 Barcelona, Spain

**Keywords:** chronic fatigue syndrome, coenzyme Q10, fibromyalgia, long COVID, mitochondrial dysfunction, myalgic encephalomyelitis, post-viral fatigue syndrome

## Abstract

Post-viral fatigue syndrome (PVFS) encompasses a wide range of complex neuroimmune disorders of unknown causes characterised by disabling post-exertional fatigue, myalgia and joint pain, cognitive impairments, unrefreshing sleep, autonomic dysfunction, and neuropsychiatric symptoms. It includes myalgic encephalomyelitis, also known as chronic fatigue syndrome (ME/CFS); fibromyalgia (FM); and more recently post-COVID-19 condition (long COVID). To date, there are no definitive clinical case criteria and no FDA-approved pharmacological therapies for PVFS. Given the current lack of effective treatments, there is a need to develop novel therapeutic strategies for these disorders. Mitochondria, the cellular organelles responsible for tissue energy production, have recently garnered attention in research into PVFS due to their crucial role in cellular bioenergetic metabolism in these conditions. The accumulating literature has identified a link between mitochondrial dysfunction and low-grade systemic inflammation in ME/CFS, FM, and long COVID. To address this issue, this article aims to critically review the evidence relating to mitochondrial dysfunction in the pathogenesis of these disorders; in particular, it aims to evaluate the effectiveness of coenzyme Q10 supplementation on chronic fatigue and pain symptoms as a novel therapeutic strategy for the treatment of PVFS.

## 1. Introduction

Post-viral fatigue syndrome (PVFS) comprises common neuroimmune conditions of unknown aetiology based on the updated WHO International Classification of Diseases for Mortality and Morbidity Statistics (https://icd.who.int/browse11/l-m/en, accessed on 15 September 2023). PVFS includes myalgic encephalomyelitis/chronic fatigue syndrome (ME/CFS), fibromyalgia (FM), and recently post-COVID-19 condition (long COVID). PVFS is characterised by prolonged post-exertional fatigue as the hallmark symptom, which worsens with minimal physical and mental exertions, along with myalgia and joint pain; cognitive impairments; unrefreshing sleep; dysautonomia; and neuropsychiatric symptoms, such as emotional lability, anxiety/depression, and apathy, which typically occur following repeated viral infections. Although PVFS is not a generally recognised disorder, it has recently become increasingly associated with the post-COVID-19 condition [1,2].

There are over 65 million people worldwide suffering from chronic disabling disorders who are diagnosed without a clear elucidation of pathophysiologic mechanisms. In addition to this massive disorder burden, the CDC now estimates that a large increase in the prevalence of ME/CFS after COVID-19 is expected to affect more than 150 million people by 2050 worldwide, a challenge still unresolved by global healthcare system [3].

To date, there are no definitive diagnostic case criteria, no accurate diagnostic tests, and no FDA-approved pharmacological treatments for PVFS. Because the exact aetiology of this disorder is not fully understood, together with the unpredictable nature of symptoms, the diagnosis and management of the disease can be challenging. The onset of disease is multifactorial (e.g., a combination of immunogenetic and environmental factors); can occur suddenly or develop gradually over time; and is typically associated with triggering events, which, in addition to viral infections, may include physical and emotional trauma [4]. PVFS is often triggered by common viral infections such as the Epstein–Barr virus (EBV), human herpesvirus (HHVs), cytomegalovirus (CMV), SARS-CoV-2 (COVID-19), amongst others [5,6].

ME/CFS, FM, and long COVID are debilitating multisystem conditions that affect most body systems and are characterised by overwhelming fatigue and widespread musculoskeletal pain that is not alleviated by rest and cannot be explained by any underlying medical condition. In addition to fatigue and chronic pain as the “prime” symptoms, individuals may experience an array of symptoms such as post-exertional malaise, unrefreshing sleep, cognitive impairments commonly known as “brain fog”, orthostatic intolerance, and gastrointestinal complaints [7,8,9].

These conditions affect millions of people of any age, gender (although predominantly women), and socio-economic burden worldwide, and the widely varying impact on individuals’ daily functioning and quality of life can be significant [10,11]. The severity of the disease can fluctuate in frequency and intensity over time among patients, ranging from mild to moderate symptoms, while others can be severely affected, with around 25% housebound or bedridden [12,13]. The above conditions are established disorders, and their background will not be described further. Long COVID is a more recently recognised disorder, and it is further described below.

Post-acute sequelae of SARS-CoV-2 infection (PASC), also known as post-COVID-19 condition or long COVID, is an emerging umbrella condition, defined as a constellation of ongoing, relapsing, or new symptoms experienced by people following an acute COVID-19 infection that continue for months and even years [14,15]. While most individuals recover from a COVID-19 infection within a few weeks, some of them continue to experience prolonged symptoms that can significantly impact their daily functioning and quality of life. People who experience long COVID sometimes refer to themselves as “long-haulers” [16]. The term “long COVID” will be used throughout this review.

A significant portion of convalescent COVID-19 patients, estimated at 10–30% (over 30 million people in the U.S., 20 million in Europe, and up to 180 million worldwide) may experience long COVID [17]. The wide variation in the estimated prevalence of long COVID within and between countries may result from a number of determinants, including the age and sex of subjects, comorbid health conditions, the timing of assessment, sociodemographic factors, self-reported questionnaire variabilities, etc. [18]. There is emerging evidence suggesting that some individuals with long COVID exhibit symptoms common to patients with ME/CFS and FM, indicating potential overlapping biological pathomechanisms, although this is still unclear [1].

The precise pathomechanism underlying long COVID is unknown and is the subject of ongoing research. Risk factors for long COVID include increasing age, obesity, pre-existing respiratory disorders, and sociodemographic factors [18]. However, it is not yet clear why some individuals develop long COVID, while others recover fully after an acute COVID-19 infection. Long COVID can affect individuals varying in the severity of initial infection, including asymptomatic children/adolescents [19]. Long-term longitudinal clinical and -omics studies are needed to determine if long COVID can lead to distinct subgroups using cluster analysis in some individuals with ME/CFS and FM [20,21].

ME/CFS, FM, and long COVID can occur independently or an coexist in some individuals with other comorbid health conditions, such as irritable bowel syndrome (IBS), and mood disorders, such as anxiety/depression, which can further complicate the understanding and management of these conditions [22]. There is currently no cure for these disorders; a multidisciplinary approach that focuses on symptom relief, pacing activities, and improving overall well-being for affected people is often employed [23,24,25,26]. While the exact cause of these illnesses remains unknown, ongoing research is crucial to unravel the complexities of the connection between these conditions, and to develop a prominent hypothesis to deepen our understanding of the aetiology, underlying pathomechanisms, and risk factors; to develop more effective diagnostic tools; and to identify more effective treatment strategies.

This article aims to evaluate the evidence relating to mitochondrial dysfunction in the pathogenesis of PVFS; in particular, it aims to review issues relating to the efficacy of coenzyme Q10 (CoQ10) supplementation as a novel therapeutic strategy for the treatment of post-viral fatigue syndrome.

## 2. Evidence of Mitochondrial Dysfunction in Post-Viral Fatigue Syndrome

Mitochondria, the cellular powerhouses responsible for energy production, have recently garnered attention in the research into PVFS due to their crucial role in cellular energy metabolism. In addition to their role in energy production, mitochondria have key roles in many other aspects of cell metabolism, including free radical metabolism, calcium homeostasis, pyrimidine and lipid synthesis, and apoptosis [27,28,29]. The accumulating literature has identified a link between mitochondrial dysfunction (including oxidative stress, redox imbalance, altered mitochondrial membrane potential/permeability, disrupted calcium homeostasis, and impaired ATP production) and low-grade systemic inflammation in ME/CFS, FM, and long COVID patients [30,31].

Research has indicated that patients with ME/CFS, FM, and long COVID often exhibit abnormalities in mitochondrial function. Studies have shown that decreased ATP production, impaired mitochondrial respiration, abnormal mitochondrial DNA levels, immune dysregulation, increased oxidative stress, imbalance redox metabolism, and chronic systemic inflammation perpetuate symptoms in these patients [32,33,34]. Additionally, abnormalities in the mitochondrial structure and function have been also observed in muscle biopsies, indicating a systemic impact on energy production. These findings suggest that mitochondrial dysfunction could contribute to the energy depletion and fatigue experienced in these disorders [35,36,37]. In addition to fatigue, these diseases are frequently accompanied by a variety of overlapping symptoms, such as cognitive impairments, sleep disturbances, brain fog, concentration/memory impairments, and muscle pain. These symptoms can be attributed, at least partially, to mitochondrial dysfunction, which can have an impact on brain energy metabolism [38,39] (Figure 1).

Several studies have reported mitochondrial abnormalities in skeletal muscle cells of ME/CFS patients compared to healthy controls. Another study reported lower mitochondrial respiration rates and increased mitochondrial reactive oxygen species production in the immune cells of ME/CFS patients. These findings suggest that there are potential mitochondrial impairments in ME/CFS, contributing to energy production deficits and oxidative stress [40,41]. Similarly, in FM, some studies have observed evidence of mitochondrial dysfunction. Research has shown reduced mitochondrial ATP production, impaired oxidative phosphorylation, and increased oxidative stress biomarkers in the muscle cells of FM patients. However, more studies are needed to validate and expand upon these findings [34,42].

Regarding post-COVID-19 syndrome, emerging research has indicated potential evidence of mitochondrial dysfunction in the blood immune cells of COVID-19 patients. When the COVID-19 virus first enters the host respiratory tract, infection is initiated via the binding of the spike protein with angiotensin-converting enzyme 2 (ACE2) receptors, with the subsequent utilization of the transmembrane protease-serine 2 (TMPRSS2) to enter host cells; the virus then hijacks the host cellular machinery for viral RNA replication and protein production [43]. In a similar manner, there is evidence that the SARS-CoV-2 is able also to hijack the host cells’ mitochondria for viral advantage, for example, to evade host immune response [44]. It is of note that many other viral, bacterial, fungal, or parasitic pathogens also modulate the host cells’ mitochondrial function to evade a host immune response and to promote infection [45]. The subject of mitochondrial hijacking in fatigue-related disorders is further described in a subsequent section of this article.

COVID-19 can cause systemic inflammation and oxidative stress, which may impact mitochondrial function. Preliminary studies have shown mitochondrial abnormalities in tissues and cells affected by COVID-19, including the lung epithelial cells and immune cells. However, more research is needed to understand the specific relationship between mitochondrial dysfunction and the post-COVID condition [46]. Several mechanisms may underlie mitochondrial dysfunction in PVFS. These include viral-induced mitochondrial damage, dysregulation of mitochondrial biogenesis and dynamics, immune-mediated mitochondrial dysfunction, and increased oxidative stress. Additionally, dysregulated mitochondrial calcium handling and impaired mitochondrial membrane potential may further contribute to the pathogenesis of PVFS [47,48,49].

One theory proposes that mitochondrial dysfunction in these diseases could be caused by a combination of genetic predisposition and environmental factors. Genetic variations in genes involved in mitochondrial function and energy metabolism, such as those related to the mitochondrial DNA, electron transport chain components, and oxidative stress response, may increase the susceptibility to mitochondrial dysfunction in individuals with PVFS. Moreover, various environmental triggers, including infectious agents, toxins, and physical or emotional stressors, have been proposed as potential triggers that can induce or exacerbate mitochondrial dysfunction in susceptible individuals. These triggers may lead to mitochondrial damage, oxidative stress, and inflammation, further impairing mitochondrial function and perpetuating the cycle of fatigue and other core symptoms in people with PVFS [50,51,52,53,54,55].

Understanding the connection between mitochondrial dysfunction and cardinal symptoms in PVFS has important clinical implications. Biomarkers of mitochondrial dysfunction, such as markers of oxidative stress and mitochondrial DNA damage, may aid in the diagnosis and subtyping of PVFS. Furthermore, targeting mitochondrial dysfunction through therapeutic interventions aimed at improving the mitochondrial function and reducing oxidative stress and imbalance redox holds promise for the management of PVFS. For example, supplementation with CoQ10 has shown promise in alleviating symptoms in some PVFS patients. However, further research is crucial to unravel the complexities of this connection, to develop targeted interventions to improve the quality of life for these individuals, and to explore the effectiveness and safety of such interventions [56,57,58,59,60,61,62,63,64]. The rationale for CoQ10 supplementation with regard to mitochondrial hijacking is considered in the following section.

## 3. Hijacking of Host Mitochondria in Post-Viral Fatigue Related Disorders

In addition to their role in cellular energy provision, mitochondria also have a key role in the host innate immune response in the first line defence against RNA viruses. Viral infection results in the activation of mitochondrial antiviral signalling proteins (MAVSs), which in turn results in the release of cytokines/chemokines and growth factors by the infected cell; this induces a further immune response which kills the infected host cell, facilitating clearance of the infecting virus [65,66]. Certain types of virus such as SARS-CoV-2 have adapted to promote viral survival and replication by suppressing the host immune response; by forming double-membrane vesicles (DMVs) around its RNA, thus shielding the latter from detection; and by inhibiting the MAVSs in the antiviral innate immune response [67]. While these DMVs are generally believed to be formed via viruses manipulating the endoplasmic reticulum (ER) membrane, the mechanism for importing and packaging proteins and RNA into these miniature organelles is not clearly understood [68]. One possible mechanism for importing viral RNA involves the virus exploiting the RNA localization mechanisms that the cell already possesses for endogenous double-membrane organelles, namely, the mitochondria.

In addition, the COVID-19 virus can directly impair the mitochondrial energy metabolism via targeted action on oxygen availability and utilization, and an effective host immune response will be impaired when the available mitochondrial energy is reduced [69]. COVID-19 viruses produce accessory proteins called open reading frames (ORFs), which interact with mitochondrial outer membrane receptors. One particular interaction involves an ORF-9 interaction with MAVSs in their role as mitochondrial import receptors and cytoplasmic viral recognition receptors [70]. Thus, ORFs have been shown to suppress the activity of MAVSs, thus limiting the initial host cell, innate immune, interferon, and antiviral responses. The interactions of the SARS-CoV-2 proteins such as ORFs and NSP with host cell mitochondrial proteins lead to the loss of membrane integrity and also cause dysfunction in the bioenergetics of the mitochondria.

The hijacking of the mitochondria by intracellular viral RNA and protein components also occurs during infections with the Ebola, Zika, and influenza A viruses [71]. It follows that drugs that help to prevent mitochondrial hijacking or restore mitochondrial function (of which CoQ10 would be an example) may provide novel therapeutic strategies to help prevent or treat the virus infection [72]. As noted in the section of this article on COVID-19 infections, to date variable outcomes have been reported from clinical studies supplementing CoQ10 in COVID-19 patients. Thus, an open study by Barletta et al. reported supplementation with 200 mg/day of CoQ10 and 200 mg/day of alpha lipoic acid for 2 months improved fatigue in chronic COVID-19 patients [73]. However, a randomised controlled trial supplementing CoQ10 (500 mg/day for 6 weeks) found no significant benefit on reducing the number or severity of symptoms in patients with the post-COVID-19 condition [74]. The development of novel therapies based on countering the effects of the viral hijacking of host mitochondria in post-infectious fatigue disorders therefore remains an area for more intensive future research.

## 4. Effects of Coenzyme Q10 Supplementation in Post-Viral Fatigue Related Conditions

All of the clinical studies described below supplemented the oxidised form of CoQ10 (ubiquinone) unless otherwise specified (Table 1).

### 4.1. Coenzyme Q10 Supplementation in Fibromyalgia

With regard to interventions through nutritional supplementation in fibromyalgia, CoQ10 (alone or in nutrient combinations) has been shown to be effective in clinical studies for the symptomatic relief of this disorder. As noted in earlier sections of this article, fibromyalgia patients have depleted CoQ10 levels in tissues (typically 40–50% of the normal level), together with increased levels of mitochondrial dysfunction, oxidative stress, and inflammation, both in adults and in children/adolescents [85,86].

A randomised controlled clinical study by Cordero et al. [75] conducted in 20 FM patients found supplementation with CoQ10 (300 mg/day for 40 days) significantly reduced (by more than 50%) chronic pain and fatigue; there was a corresponding improvement in mitochondrial energy generation and reduced oxidative stress and inflammation. In this study, psychopathological symptoms (including anxiety/depression) were also significantly improved; this was linked to the effect of supplemental CoQ10 in reducing oxidative stress and inflammation, as well as increased levels of serotonin [76,77]. In addition, Cordero et al. [78] correlated headache symptoms with reduced CoQ10 levels and increased oxidative stress in the FM patients, with headache symptoms and oxidative stress levels significantly improved following CoQ10 supplementation (300 mg/day for 3 months). In adolescent FM patients (aged 8–17 years), Miyamae et al. [86] reported CoQ10 supplementation (100 mg/day for 3 months) significantly improved fatigue. In an open-label crossover study, supplementation with a water-soluble form of CoQ10 (400 mg/day for 6 months) resulted in significant reductions (by 20–40%) in chronic fatigue and pain [79].

A clinical study by Sawaddiruk et al. [80] described the role of supplementary CoQ10 in further reducing chronic pain in FM patients treated with pregabalin. In a double-blind randomised controlled study, the FM patients were treated either with pregabalin plus CoQ10 or pregabalin plus a placebo for 6 weeks. After a 2-week washout period, patients in the pregabalin plus CoQ10 group were switched to pregabalin plus a placebo, and vice-versa, for a further 6 weeks. Several parameters were monitored during the study, including the pain pressure threshold, pain score, and anxiety/depression level, as well as the biomarkers of antioxidant activity, inflammation, and mitochondrial dysfunction. While pregabalin alone reduced pain and anxiety/depression, there was no effect on inflammation and mitochondrial function. However, the treatment with pregabalin plus CoQ10 resulted in a significantly greater reduction in pain and anxiety/depression, together with a reduction in oxidative stress, inflammation, and mitochondrial dysfunction. The results from this study provide evidence that supplementary CoQ10 can provide further pain relief in FM patients treated with pregabalin, through reducing oxidative stress imbalance and inflammation and improving mitochondrial function.

With regard to nutrient combinations, an open uncontrolled study of 23 FM patients reported supplementation with 200 mg of CoQ10 and 200 mg of ginkgo biloba resulted in a significant improvement in health-related quality of life [81]. In a randomised controlled trial conducted in 21 FM patients supplemented with CoQ10, vitamin D, alpha-lipoic acid, magnesium, and tryptophan for 3 months, there was a significant reduction in pain [82].

### 4.2. Coenzyme Q10 Supplementation in Acute COVID-19 Infection

In a clinical study by Israel et al. [83], the intake of CoQ10 was associated with a significantly reduced risk of hospitalisation for COVID-19 patients. Fernandez-Ayala et al. [87] reviewed the evidence for mitochondrial dysfunction as a key factor determining the severity of a SARS-CoV-2 infection; in particular, the authors noted the increased susceptibility to a COVID-19 infection in individuals over 65 years of age, the same age by which endogenous CoQ10 levels have become substantially depleted. Similarly, Gvozdjakova et al. [72] consider one of the main consequences of a COVID-19 infection to be virus-induced oxidative stress-causing mutations in one or more of the genes responsible for CoQ10 biosynthesis, in turn resulting in mitochondrial dysfunction. Also of note is the computational study by Caruso et al. [88], in which the authors identified CoQ10 as a compound capable of inhibiting the SARS-CoV-2 virus, via binding to the active site of the main viral protease.

Clinical studies supplementing CoQ10 in COVID-19 patients (and also patients with long COVID) have reported mixed outcomes. In a prospective observational study, 116 patients with chronic COVID-19 were supplemented with 200 mg of CoQ10 and 200 mg of alpha-lipoic acid for 2 months versus 58 COVID-19 patients who received no treatment; the severity of fatigue was substantially reduced in the treated patients compared to the placebo group [73]. However, a randomised controlled intervention study comprising 121 chronic COVID-19 patients supplemented with 500 mg of CoQ10 per day for 6 weeks reported no significant benefit on chronic COVID-19 symptoms [74].

In patients severely affected by a COVID-19 infection, in addition to the “cytokine storm” and hyperinflammation status affecting the lung tissue, cardiac injury biomarkers may be elevated, characteristic of myocarditis and heart failure. While the precise mechanism underlying the damaging effect of COVID-19 infections on the heart function is not completely understood, attention has been focussed on the ACE-2 enzyme, which has roles both in cardiovascular function and the penetration of host cells by the SARS-CoV-2 infection.

It is well known that people with pre-existing medical conditions such as cardiovascular disease are at a greater risk of adverse outcomes following a COVID-19 infection. The acute effects of a COVID-19 infection in terms of cardiovascular function are well established, as noted above. What is less well known is the risk of longer-term cardiovascular dysfunction following a COVID-19 infection. Individuals who have been hospitalised because of a COVID-19 infection have a much higher risk of subsequently developing heart problems, including myocarditis, atrial fibrillation, myocardial infarction, heart failure, and stroke. However, it is not just people who have suffered more serious COVID-19 infections who are prone to developing heart problems; even people who were not hospitalised and seemed to have recovered from mild infection were subsequently found to be at risk of developing serious heart complications.

These conclusions were based on a number of clinical studies published in the medical literature, including studies led by Dr Ziad Al-Aly (Washington University Medical School, St. Louis, MO, USA), Dr Mouaz Al-Mallah (Debakey Heart Centre, Houston, TX, USA), and Prof. Colin Berry (Glasgow University). Dr Al-Aly’s study comprising some 150,000 patients found both hospitalised and non-hospitalised individuals were at a substantially increased risk of developing a number of heart conditions, including myocarditis and heart failure [89]. The study by Dr Al-Mallah of 100 COVID-19 patients assessed via positron emission tomography/myocardial blood flow found that an infection doubled the risk of developing unhealthy endothelial cells lining the heart and blood vessels, increasing the likelihood of heart failure [90]. Additionally, Prof Berry’s study, comprising more than 1000 patients, found that one in eight individuals who had been hospitalised due to COVID infection were later diagnosed with myocarditis [91]. The pathophysiological mechanisms by which SARS-CoV-2 infection causes longer term cardiovascular dysfunction is not completely understood. Possible contributory factors include direct cellular damage resulting from the viral invasion of cardiomyocytes and subsequent cell death, endothelial cell infection and endothelitis, persistent hyperactivation of the immune response, and persistence of the virus in tissues, possibly lying dormant for months or even years to be reactivated under conditions of stress/insults.

The above data provide a rationale for the use of supplemental CoQ10 to try and prevent the development of heart problems following COVID-19 infection; in addition, there is evidence that supplemental selenium may also be of benefit, as outlined below. There are several reasons why the nutritional supplements CoQ10 and selenium could be of benefit regarding COVID infection. Firstly, both CoQ10 and selenium have important roles in immune function and could help to prevent a COVID infection from taking place. Secondly, the combination of CoQ10 and selenium can reduce the excessive inflammation associated with virus infections, as well as inflammation in individuals without virus infections. Thirdly, CoQ10 and selenium have important roles in the normal function of the heart, and clinical trials supplementing CoQ10 and selenium have shown significant benefit in reducing the risk of developing heart disease in normal individuals as well as reducing mortality risk in heart failure patients [92,93,94,95]. It should be noted that the use of CoQ10 and selenium described above is a suggestion of the authors of this article and has not been explored in clinical trials to date, but an open-label exploratory study has been conducted to evaluate the efficacy of combined CoQ10 and selenium supplementation on clinical features and circulating biomarkers in ME/CFS patients [64]. Based on these findings, long-term supplementation with CoQ10 plus selenium suggests a potentially beneficial synergistic effect in patients with ME/CFS.

### 4.3. Coenzyme Q10 Supplementation in Myalgic Encephalomyelitis/Chronic Fatigue Syndrome

In a randomised controlled trial comprising 207 ME/CFS patients who were supplemented with 200 mg of CoQ10 and 20 mg of NADH per day for 3 months, or a placebo, there was a significant improvement in fatigue perception (assessed on the FIS-40 scale), health-related quality of life (36-item, short-form health survey) and sleep quality (assessed by a PSQI questionnaire) [63]. In an open-label study, 27 ME/CFS patients were supplemented with 400 mg of CoQ10 and 200 mcg of selenium per day for 8 weeks; there was a significant improvement in overall fatigue severity and global quality of life compared to the baseline among the participants [64]. Supplementation with CoQ10 for symptoms such as fatigue, pain, and cognitive dysfunction in ME/CFS has been suggested in an agreed report from the European Network on ME/CFS (EUROMENE consortium) [96]. In a randomised controlled trial, 43 ME/CFS patients were supplemented with the reduced CoQ10 form (ubiquinol) (150 mg/day for 3 months) or a placebo. Although there was no improvement in perceived fatigue (assessed by Chandler’s fatigue scale), supplementation improved several other illness symptoms such as night-time awakenings [84].

A potential issue regarding the success or otherwise of clinical trials is the quality and bioavailability of the CoQ10 supplement used, and this is addressed in the following section.

## 5. Importance of CoQ10 Supplement Quality and Bioavailability

All prescription-type drugs require a marketing authorisation. To obtain a marketing authorisation, manufacturers must submit to the relevant regulatory authorities an extensive dossier of data relating to product manufacturing quality, together with proof of efficacy and safety [97]. Once marketing authorisation has been approved, products must be subject to a continual process of pharmacovigilance [98]. By contrast, products classed as food supplements are not subject to the same regulatory standards; there is no mandatory requirement relating to product quality or proof of efficacy and safety. As an example, surveys of Ginkgo biloba supplements available on the European and the US markets have reported issues with formulation discrepancies and adulteration problems. There is similarly no regulatory requirement for the manufacturers of most CoQ10 supplements to guarantee the quality, efficacy, and safety of their products [99,100,101].

To date, there is only one CoQ10 product produced to pharmaceutical standards for the adjunctive treatment of heart failure. Marketing authorisation approval acts as a guarantee relating to the accuracy of the stated dosage, the absence of adulterants, and their bioavailability. Products manufactured according to food supplement standards do not have this guarantee, which in turn may have adverse effects on the outcome of clinical studies utilising such products. In this regard, a recent review by Mantle & Hargreaves identified 38 clinical studies in which supplemental CoQ10 had been used to treat patients with primary CoQ10 deficiency. Only two of these studies provided information about the manufacturer of the supplement used. Based on this limited information, it becomes difficult to evaluate the outcomes of such studies in which poor-quality supplements may have been used [102].

The other key issue relevant to the success or otherwise of clinical studies is that of bioavailability. Bioavailability is defined as the proportion of an ingested substance that reaches the bloodstream. CoQ10 is a lipid-type substance and as such is absorbed from the digestive tract in the same general manner as other lipid substances. The process by which this takes place has been described in detail in the review by Mantle & Dybring [103] from which the following information has been summarised. Because of the particular chemical structure of CoQ10 (one of the most hydrophobic naturally occurring substances), the bioavailability of supplemental CoQ10 is low, estimated to be around 5% at most. One of the most effective methods to date for optimising the bioavailability of CoQ10 is based on a patented CoQ10 crystal modification process used in the manufacture of ubiquinone-form CoQ10 supplements [103].

Coenzyme Q10 is produced via a yeast fermentation process in the form of polymorphic crystals, which cannot be absorbed from the digestive tract. CoQ10 can be absorbed only as individual molecules, as noted above. To be effective as a supplement, the CoQ10 crystals must therefore be dissociated first into individual CoQ10 molecules prior to absorption. The above process involves changing the shape of the CoQ10 crystals in such a way as to increase the ratio of the crystals’ surface area to the volume, thus making it easier for the crystals to dissolve into single molecules at body temperature. This modification to the CoQ10 crystalline form should remain in place throughout the shelf life of the CoQ10 preparation.

Supplement manufacturers may make extravagant claims about the bioavailability of their respective CoQ10 supplements, but the only definitive measure of bioavailability is that determined in human subjects, based on clinical studies published in the peer-reviewed medical literature. A good example in this regard is the bioavailability study by Lopez-Lluch et al. [104]. In this randomised controlled clinical trial, seven CoQ10 supplements differing in formulation (the CoQ10 crystal modification status, the type of carrier oil, the composition of other excipients, and the CoQ10 oxidation state) were administered in a single 100 mg dose to the same series of 14 healthy individuals, using a crossover/washout protocol. The bioavailability of the different formulations was quantified as the area under the curve at 48 h. The supplement that had been subject to the crystal modification process (Myoqinon) had the highest level of bioavailability, whilst the bioavailability of the same CoQ10 material that had not been subject to this process was reduced by 75%. It is of note that the Myoqinon formulation received a marketing authorisation within the E.U., as noted above, demonstrating the importance of utilising a CoQ10 product manufactured to pharmaceutical standards, rather than to food supplement standards.

Several studies have been carried out with the objective of improving the bioavailability of CoQ10 using a variety of agents; examples include polyethylene glycol, phosphorylated tocopherols, poloxamer/polyvinyl pyrrolidine, and hydrolysed proteins. However, again, the bioavailability of most of these formulations has not been directly compared with ubiquinone that has undergone crystal modification, the importance of which is demonstrated in the above Lopez-Lluch et al. study [104]. In addition, none of the modified forms of CoQ10 described above have been subject to an extensive evaluation of their efficacy and safety in randomised controlled trials. In comparison, the efficacy and the safety of the crystal-modified form of CoQ10 have been confirmed in a number of such clinical studies, an example being the Q-SYMBIO study in which supplemental CoQ10 was shown to substantially reduce the mortality risk in heart failure patients [95].

Notwithstanding the issues outlined above, it should be noted that there are other factors that may influence the success of clinical studies involving CoQ10 supplementation, as reviewed by Mantle et al. [105]. Examples include the problem that some individuals appear to have an inherently low capacity to absorb supplemental CoQ10 into the bloodstream, even with high bioavailability formulations—the reason for which is currently unknown. Additionally, the question of whether supplemental CoQ10 can effectively cross the blood–brain barrier in human subjects remains unclear.

## 6. Conclusions, Unresolved Issues, and Future Perspectives

Disorders constituting PVFS, namely, ME/CFS, FM, and long COVID, currently have no effective treatments. As part of a strategy to identify new treatment methods, we have identified considerable evidence for the involvement of mitochondrial dysfunction in the pathogenesis of these disorders. This, in turn, suggests a possible role for CoQ10 supplementation in the treatment of the above disorders, given the key role of CoQ10 in the normal mitochondrial function. Within mitochondria, CoQ10 has a key role as an electron carrier (from complexes I and II to complex III) in the mitochondrial electron transport chain during oxidative phosphorylation to generate ATP. In addition, CoQ10 serves as an important lipid-soluble antioxidant protecting the mitochondria from oxidative stress-induced damage resulting from reactive oxygen/nitrogen free radical species generated during oxidative phosphorylation. CoQ10 is also involved in the metabolism of pyrimidines, fatty acids, and mitochondrial uncoupling proteins, as well as in the regulation of the mitochondrial permeability transition pore [106]. Therefore, CoQ10 supplementation may ameliorate mitochondrial dysfunction and alleviate fatigue associated with PVFS through its ability to restore electron flow in the mitochondrial respiratory chain (MRC), which will enhance ATP generation by oxidative phosphorylation. Furthermore, exogenous CoQ10 will also enhance mitochondrial antioxidant capacity, which will protect the enzymes of the MRC from reactive oxygen species-induced impairment [107]. Interestingly, several clinical studies have demonstrated an association between decreased endogenous CoQ10 levels and an increased susceptibility to COVID-19 infection, and, consequently, some patients experiencing post-viral fatigue may have an underlying CoQ10 deficiency, which CoQ10 supplementation may restore [108].

Randomised controlled clinical trials have reported significant symptomatic benefits in the treatment of these disorders, particularly for FM; further randomised controlled trials are required to confirm the efficacy of CoQ10 supplementation in patients with ME/CFS and long COVID. It is of note that no serious adverse effects have been reported from clinical studies supplementing CoQ10 in patients with PVFS, or in a wide range of other chronic disorders. Other potential therapeutic agents for the treatment of PVFS have been suggested (reviewed by Chen et al. [109]); these include generalised antioxidants which can restore mitochondrial function, such as N-acetylcysteine; glutathione and catalase; IL-6R and IL-1 receptor blockers, such as the antibody tocilizumab, to reduce levels of pro-inflammatory cytokines; and mitochondrial calcium uniporter (MCU) inhibitors, such as ruthenium-265, mitoxantrone, and doxycycline, which can help restore mitochondrial function. However, in contrast to the use of CoQ10, none of the above agents have to date been investigated in randomised controlled trials for PVFS.

There are several unresolved issues relating to the CoQ10 intervention studies and mitochondrial dysfunction in PVFS that require further investigation; these include whether the bioavailability of CoQ10 could be improved through the use of alternative administration routes (e.g., intravenous, intraperitoneal, intramuscular, etc.), whether supplemental CoQ10 is able to cross the blood–brain barrier, and how CoQ10 is transported into and within cells, among others. With regard to future work, the question arises whether additional symptomatic benefits may be obtained through co-supplementation with other substances that have important roles in the mitochondrial function, as suggested by Castro-Marrero et al. and Mantle and Hargreaves [63,110]. The possibility of designing dietary interventions based on CoQ10 co-supplementation targeted specifically at boosting mitochondrial function to improve neuroimmune and inflammatory health outcomes in ME/CFS, FM, and long COVID appears well within reach within the next half decade. Mitochondria-dependent pathways may thereby represent an attractive therapeutic target for the amelioration of PVFS.

## Figures and Tables

**Figure 1 ijms-25-00574-f001:**
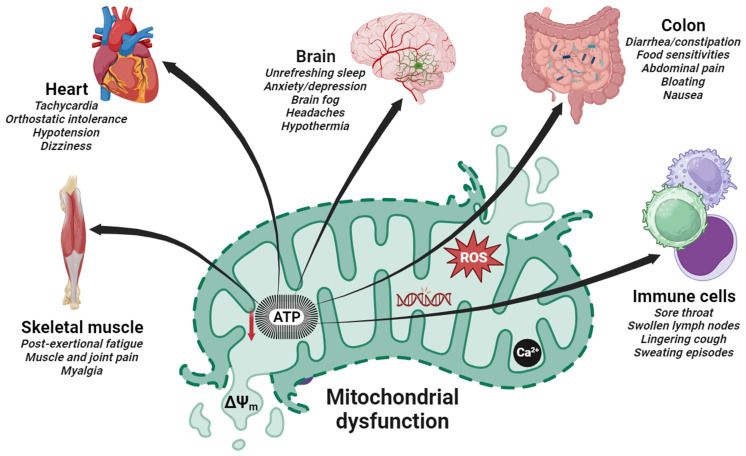
Schematic representation of mitochondrial dysfunction involved in the pathogenesis of post-viral fatigue syndrome. The most prevalent clinical features in these disorders affect the brain, the heart, skeletal muscle cells, immune cells, and the intestine (colon). Dysfunctions are commonly caused in multiple organ systems with a clinical spectrum that varies within and between patients. The resulting fall in ATP production due to a reduced mitochondrial membrane potential (ΔΨ_m_) and an increased ROS generation has a deleterious effect on a number of major ATP-consuming organs in patients with PVFS causing the cardinal symptoms shown.

**Table 1 ijms-25-00574-t001:** Summary of clinical trials conducted on mitochondria dysfunction-targeted CoQ10 supplementation in post-viral fatigue syndrome.

Population	Case Criteria	CoQ10 Daily Dose and Duration	Sample Size	Study Design	Outcomes	Refs.
FM	1990 ACR	300 mg for 40 days	20	RCT	Reduced chronic pain and fatigue, improved mitochondrial bioenergetic function, and reduced oxidative stress and inflammation	Cordero et al., 2013 [75]
FM	2010 ACR	300 mg for 40 days	20	RCT	Regulated serotonin levels in platelets and improved depression symptoms	Alcocer-Gómez et al., 2014 and 2017 [76,77]
FM	1990 ACR	300 mg for 40 days	20 cases 15 controls	Case-control study	Significant negative correlations between CoQ10 or catalase levels in PBMCs and headache parameters, restored biochemical parameters, and improved clinical symptoms	Cordero et al., 2012 [78]
FM	1990 ACR	2 × 200 mg for 6 months	22	Open-label crossover study	Significantly improved most pain-related outcomes by 24–37%, including fatigue (by ~22%) and sleep disturbance (by ~33%)	Di Pierro et al., 2017 [79]
FM	2010 ACR	Pregabalin with CoQ10 or pregabalin with placebo for 40 days	11	RCT crossover study	Reduced greater pain, anxiety, inflammation, and mitochondrial oxidative stress, along with increased GSH levels and superoxide dismutase levels	Sawaddiruk et al., 2019 [80]
FM	Not given	200 mg CoQ10 and 200 mg Ginkgo biloba extract for 84 days	25	Open-label pilot study	Improved quality of life and improved self-rating with 64% claiming to be better and only 9% claiming to feel worse	Lister et al., 2002 [81]
FM	2016 ACR	150 mg CoQ10 combined with vitamin D, alpha-lipoic acid, magnesium, and tryptophan or acupuncture treatment, both for 3 months	55	RCT	Reduced pain at 1 month after the start of therapy, strengthened after 3 months with the maintenance of treatment	Schweiger et al., 2020 [82]
Long COVID	eHRs from Israel Clalit Health Services provider	Not given	6953 cases 6530 controls	Retrospective case-control study	Case studies showing ubiquinone associated with significantly reduced odds for COVID-19 hospitalization (OR = 0.185, 95% CI (0.058–0.458), *p* < 0.001)	Israel et al., 2021 [83]
Long COVID	2015 IOM/NIH criteria for ME/CFS	100 mg CoQ10 and 100 mg alpha-lipoic acid (*n* = 116) or placebo (*n* = 58) for 60 days	174	Prospective observational study	Complete Fatigue Severity Scale response was reached more frequently in treatment group (53.5%) than in placebo (3.5%)	Barletta et al., 2023 [73]
Long COVID	2021 WHO clinical case definition	500 mg or placebo for 6 weeks, with crossover treatment after a 4-week washout period	119	RCT crossover trial	No significant benefit on chronic COVID-19 symptoms	Hansen et al., 2023 [74]
ME/CFS	1994 CDC/Fukuda	200 mg CoQ10 and 20 mg of NADH (*n* = 104) or placebo (*n* = 103) for 12 weeks	207	RCT	Reduced cognitive fatigue perception and overall FIS-40 score and improved quality of life; improved sleep duration at 4 weeks and improved habitual sleep efficiency at 8 weeks	Castro-Marrero et al., 2021 [63]
ME/CFS	1994 CDC/Fukuda	400 mg CoQ10 and 200 mcg selenium for 8 weeks	27	Open-label exploratory study	Improved overall fatigue severity and global quality of life but no significant effect on sleep disturbances; increased total antioxidant capacity, reduced lipoperoxide levels, and decreased circulating cytokine levels	Castro-Marrero et al., 2022 [64]
ME/CFS	1994 CDC/Fukuda	150 mg ubiquinol or placebo for 3 months	20	RCT	Improved autonomic function and cognitive function	Fukuda et al., 2016 [84]

## Data Availability

Not applicable.

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
