# Peer review of "Mitochondrial Dysfunction and Coenzyme Q10 Supplementation in Post-Viral Fatigue Syndrome: An Overview"

_ijms, 2024, doi:10.3390/ijms25010574_

Round 1
Reviewer 1 Report
Comments and Suggestions for Authors
This review purports to provide an overview of the use of coenzyme CoQ10 in the treatment of ME/CFS, FM and long covid. After a good introduction, the authors discuss mitochondrial dysfunction in post-viral syndrome before going on to list various clinical trials where patients have been treated with CoQ10 alone or in combination with other drugs/supplements. The next section addresses the important issue of quality control/variability in sources of CoQ10.
Overall, this is a somewhat interesting review that veers towards advertising, especially in the section on quality control. The key points could be made here without belaboring the specific benefits of a Pharma Nord product. Also, I would like to see more discussion of the potential mechanistic rationale/basis for how CoQ10 may counter mitochondrial dysfunction and consequently improve symptoms of fatigue in patients suffering from post-viral fatigue syndrome. Finally, the paragraph on prophylactic use of CoQ10 to prevent heart problems, viral infection and inflammation seems highly speculative and should at minimum be balanced with a discussion of potential safety risks.
Comments on the Quality of English LanguageThe english is mostly fine, but an additional round of proof reading is warranted.
Author Response
1- This review proposes to provide an overview of the use of CoQ10 in the treatment of ME/CFS, FM, and long COVID. After a good introduction, the authors discuss mitochondrial dysfunction in PVFS before going on to list various clinical trials where patients have been treated with CoQ10 alone or in combination with other drugs/supplements. The next section addresses the important issue of quality control/variability in sources of CoQ10.
Thank you for your positive feedback.
2- Overall, this is a somewhat interesting review that veers towards advertising, especially in the section on quality control. The key points could be made here without belaboring the specific benefits of a Pharma Nord product. Also, I would like to see more discussion of the potential mechanistic rationale/basis for how CoQ10 may counter mitochondrial dysfunction and consequently improve symptoms of fatigue in patients suffering from PVFS. Finally, the paragraph on the prophylactic use of CoQ10 to prevent heart problems, viral infection, and inflammation seems highly speculative and should at minimum be balanced with a discussion of potential safety risks.
Thank you for your suggestions. All these critical points have been adequately addressed throughout the revised manuscript. We have removed all references to Pharma Nord products from the text. Also, it has been added to further describe how CoQ10 may boost mitochondrial dysfunction and consequently improve fatigue symptoms in PVFS. Finally, we have added a note that the prophylactic use of CoQ10 and selenium to prevent CVD and PVFS - a suggestion by the authors of this article and has not been the subject of RCTs to date.
Reviewer 2 Report
Comments and Suggestions for Authors
The manuscript provides a comprehensive overview of mitochondrial dysfunction in Post-Viral Fatigue Syndrome (PVFS) disorders, particularly focusing on ME/CFS, FM, and Long COVID. The authors explore the potential therapeutic role of Coenzyme Q10 (CoQ10) supplementation, highlighting both its promising aspects and unresolved issues. Here are some suggestions for improvement:
· The introduction could be more streamlined by presenting the burden of Post-Viral Fatigue Syndrome (PVFS), its determinants, possible underlying mechanisms (acknowledging how these theories do not fully explain the phenomenon), and then subsequently, mitochondrial dysfunction can be introduced as an alternative or complementary theory.
· While the manuscript touches on Coenzyme Q10 (CoQ10) bioavailability and the crystal modification process, providing more mechanistic insights into how CoQ10 functions at the cellular and molecular levels would enhance the reader's understanding.
· In addition to CoQ10, the manuscript should also acknowledge several other novel treatments currently being tested, such as microRNA 2392 (miR-2392), generalized antioxidants like N-acetylcysteine, glutathione, and catalase, as well as IL-6R and IL-1 receptor blockers. Notably, Ruthenium red, a mitochondrial calcium uniporter (MCU) inhibitor, has also shown promise in normalizing mitochondrial morphology. Other known MCU inhibitors include ruthenium 265, mitoxantrone, and the antibiotic doxycycline. While the focus remains on CoQ10, it is essential to acknowledge these alternative strategies and justify why CoQ10 is a primary focus.
Author Response
1- The introduction could be more streamlined by presenting the burden of PVFS, its determinants, and possible underlying mechanisms (acknowledging how these theories do not fully explain the phenomenon), and subsequently, mitochondrial dysfunction can be introduced as an alternative or complementary theory.
We have noted the suggestion by the reviewer to streamline the Introduction; however, as Reviewer #1 commented that the Introduction was good, we wish to keep it in its present format.
2- While the manuscript touches on CoQ10 bioavailability and the crystal modification process, providing more mechanistic insights into how CoQ10 functions at the cellular and molecular levels would enhance the reader's understanding.
Thank you for your feedback. A note has been added on the function and underlying pathomechanisms of CoQ10 at the cellular and molecular levels.
3- In addition to CoQ10, the manuscript should also acknowledge several other novel treatments currently being tested, such as miRNA 2392 (miR-2392), generalized antioxidants like N-acetylcysteine, glutathione, and catalase, as well as IL-6R and IL-1R blockers. Notably, Ruthenium red, a mitochondrial calcium uniporter (MCU) inhibitor, has also shown promise in normalizing mitochondrial morphology. Other known MCU inhibitors include ruthenium 265, mitoxantrone, and doxycycline. While the focus remains on CoQ10, it is essential to acknowledge these alternative strategies and justify why CoQ10 is a primary focus.
Thank you for your recommendations. A note has been added on novel alternative therapeutic strategies currently in progress and also possible adverse effects of CoQ10 throughout the updated manuscript.
Reviewer 3 Report
Comments and Suggestions for Authors
Thank you for your manuscript. i think the theme is very interesting.
In my opinion this article should be changed in a systematic review to offer a good level of evidence
I have several concerns about advicing a specific company product, i think it is not acceptable.
I have also several doubts about this sentence "ME/CFS and FM are multifaceted post-viral syndromes". i do not think it is correct.
Comments on the Quality of English Languagefine
Author Response
1- In my opinion this article should be changed in a systematic review to offer a good level of evidence.
Given the relatively small number of RCTs on the effects of CoQ10 in PVFS, we did not think it appropriate to undertake a systematic review.
2- I have several concerns about advertising a specific company product, I think it is not acceptable.
As noted above, references to Pharma Nord products have been removed from the text.
3- I have also several doubts about this sentence "ME/CFS and FM are multifaceted post-viral syndromes". I don’t think it is correct.
This sentence has been modified as follows: ME/CFS, FM, and long COVID are debilitating multisystem conditions that affect most body systems.
Round 2
Reviewer 1 Report
Comments and Suggestions for Authors
This is a much improved version of the manuscript and I appreciate the authors responsiveness to my previous comments.
Reviewer 2 Report
Comments and Suggestions for Authors
All my comments have been addressed. No further comments
Reviewer 3 Report
Comments and Suggestions for Authors
I appreciate the improvement, but according to me they stduy needs to be deepen using a systematic approach.